# Role of community health workers in improving cost efficiency in an active case finding tuberculosis programme: an operational research study from rural Bihar, India

Tushar Garg [ID],[1] Manish Bhardwaj [ID],[2] Sarang Deo [ID] [3]

[1]Research, Innovators In Health, Patna, Bihar, India
[2]Operations, Innovators In Health, Patna, Bihar, India
[3]Operations Management, Indian School of Business, Hyderabad, Telangana, India

**Correspondence to**
Dr Tushar Garg;
tgarg@innovatorsinhealth.org

## ABSTRACT

**Objectives** Cost-efficient active case finding (ACF) approaches are needed for their large-scale adoption in national tuberculosis (TB) programmes. Our aim was to assess if community health workers' (CHW) knowledge about families' health status can improve the cost efficiency of the ACF programme without adversely affecting the delivery of other health services for which they are responsible.

**Design** Quasi-experimental design.

**Interventions** We evaluated an ACF programme in the Samastipur district in Bihar, India, between July 2017 and June 2018. CHWs called Accredited Social Health Activists generated referrals of individuals at risk of TB and conducted symptom-based screening to identify patients with presumptive TB. They also helped them undergo testing and provided treatment support for confirmed TB cases.

**Primary and secondary outcome measures** We compared the notification rate from the intervention region with that from a control region in the same district with similar characteristics. We analysed operational data to calculate the cost per TB case diagnosed. We used routine programmatic data from the public health system to estimate the impact on other services provided by CHWs.

**Findings** CHWs identified 9895 patients with presumptive TB. Of these, 5864 patients were tested for TB, and 1236 were confirmed as TB cases. Annual public case notification rate increased sharply in the intervention region from 45.8 to 105.8 per 100 000 population, whereas it decreased from 50.7 to 45.3 in the control region. There was no practically or statistically significant impact on other output indicators of the CHWs, such as institutional deliveries (−0.04%). The overall cost of the intervention was about US$134 per diagnosed case. Main cost drivers were human resources, and commodities (drugs and diagnostics), which contributed 37.4% and 32.5% of the cost, respectively.

**Conclusions** ACF programmes that use existing CHWs in the health system are feasible, cost efficient and do not adversely affect other healthcare services delivered by CHWs.

### Strengths and limitations of this study

► A pragmatic active case-finding implementation that used existing community health workers in the health system.
► Used a comparable control region to obtain the incremental effect of the intervention.
► Purposively selected areas, hence, not a randomised control trial.
► Patient costs incurred or averted and the national tuberculosis programme costs not included.
► The small scale of the study and geographical location limit generalisability.

## INTRODUCTION

WHO estimates about 10 million people were falling ill with tuberculosis (TB) and nearly 1.5 million dying of it in 2018.[1] Despite the continuous increase in case notifications in recent years, the 2018 estimates predict a gap of as much as 30% between the incident and notified cases globally. Progress towards WHO's target of a 90% reduction in TB incidence rate by 2035 is severely limited by existing passive case-finding approaches that wait for patients to seek care at a health facility.[2–4] As a result, these approaches fail to address significant barriers in accessing care, such as poor geographical and financial access, stigma and poor awareness.[5]

Active case finding (ACF) can address these challenges by finding previously undetected cases and promptly initiating treatment.[4 6 7] Modelling studies estimate that such strategies can decrease TB incidence.[8 9] In contrast with passive approaches, ACF is a health system initiated screening process that uses context-specific diagnostic algorithms and accommodates various implementation strategies, including mass radiography, contact investigation and house-to-house

surveys.[6 10 11] Although modelling studies have shown ACF strategies to be cost-effective, cost per diagnosed case of such programmes can be very high, thereby limiting their large-scale adoption.[8 12–14] As a result, there is limited empirical evidence from high-burden and resource-constrained settings to inform key operational decisions regarding ACF programmes: who will conduct ACF activities, how will they be integrated within the health system and how will these additional activities impact other health services.[15]

In this study, we address these questions with evidence from a novel intervention in rural India that leveraged existing community health workers (CHWs) in the public health system for ACF activities. In particular, our aim is to assess if CHWs' knowledge about health statuses of families can improve the cost efficiency of the ACF programme without adversely affecting the delivery of other health services for which they are responsible.

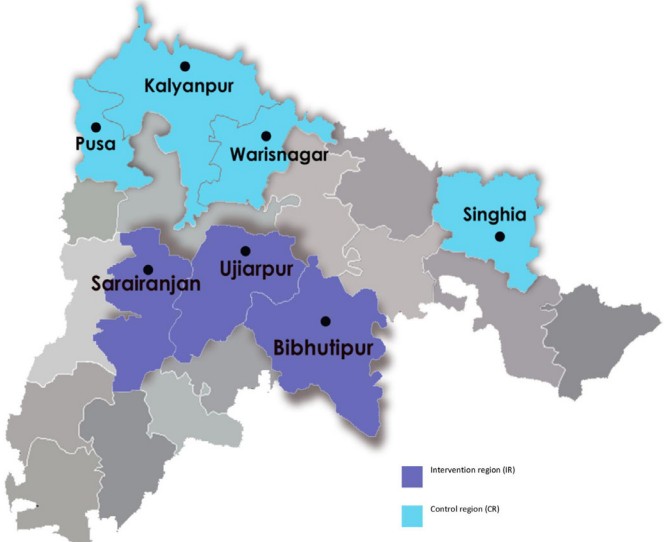

**Figure 1** Map indicating the blocks in intervention and control region in Samastipur district, Bihar.

## METHODS
### Study design
Our intervention was implemented from 15 May 2017, with state and district health administration's approval as an extension of routine services provided as part of the Revised National TB Control Programme (RNTCP). We used a quasi-experimental design to evaluate the impact of the intervention over a period between 1 July 2017 and 30 June 2018. We used the period between 15 May 2017 and 1 July 2017 in preparatory activities to launch the intervention.

### Study setting
Our study was conducted in the Samastipur district of the East Indian state of Bihar. In 2011, it had a population of about 42.6 million, of which 96.5% lived in rural areas. The literacy rate was 50.3%, and the sex ratio was 911.[16] The main source of income in more than 71.3% households was casual labour, and the highest individual income was less than INR5000 (US$71) in 69.1% of households.[17] The total fertility rate in the district was 3.8, and the infant mortality rate was 53 deaths per 1000 live births.[18] More than 70% of births occurred at a healthcare institution.[19] In 2017, the annual TB case notification rate for the district was 55 per 100 000 population with a pretreatment lost to follow-up (PTLFU) rate of 25%. In 2016, a successful treatment outcome was reported for 72% of the microbiologically confirmed (Bac+) new TB cases (44% of all cases).[20]

The intervention region (IR) consisted of three blocks—Ujiarpur, Bibhutipur and Sarairanjan—with a total population of 1 021 483.(figure 1). We chose four blocks—Kalyanpur, Warisnagar, Pusa and Singhia—as the control region (CR) with a population of 981 924.[21] The choice of these blocks was purposive with an emphasis on a similar population, sociodemographic and health system characteristics, and TB epidemiology. These were

geographically separated from the IR to minimise the spill-over effects of the intervention.

IR and CR were similar along relevant sociodemographic variables such as the proportion of the population belonging to scheduled castes (18.2% vs 20.8%)[17] (table 1). Further, the structure of the public health systems in IR and CR was comparable on relevant dimensions. Each block in IR, as well as CR, coincided with a TB unit under the RNTCP, which was managed by a senior treatment supervisor (STS). IR and CR included four designated microscopy centres each, where sputum microscopy was provided. Finally, the annual TB case notification rate was comparable across IR and CR (52 vs 53.1 per 100 000 population in 2016).[22]

### Intervention
We implemented an ACF intervention with the support of RNTCP and the National Health Mission (NHM) and project funding from Stop TB Partnership's TB REACH. Under this intervention, we engaged with CHWs, locally known as Accredited Social Health Activists (ASHAs), who work for the NHM. Their main role was community mobilisation and facilitating last-mile delivery of health services across multiple programmes though their focus is reproductive, maternal and child health. Although ASHAs were chosen from literate women between 25 and 45 years of age with a preference to those educated up to the tenth standard, the criteria were relaxed if no such woman was available in the village.[23] They received performance-linked and activity-linked remuneration, for example, US$0.7 to report a newborn death within 24 hours, US$2 to attend review meetings, US$8 for antenatal care (ANC) and institutional delivery, up to US$15 for promoting contraception, and up to US$75 supporting TB treatment (US$15 for a new case, US$22 for a previously diagnosed case and US$75 for a drug-resistant TB case).[24 25] They were supervised by ASHA

**Table 1** The demographic characteristics of the intervention and control region in the active case-finding project

| Characteristics | Intervention region | Control region |
|---|---|---|
| Blocks | 3 | 4 |
| Area (sq. km.) | 582 | 623 |
| Population | 1 021 483 | 981 924 |
| Sex ratio | 918 | 919 |
| Proportion of scheduled castes population | 18.2% | 20.8% |
| Literacy rate | 63.5% | 59.8% |
| Households with monthly income of highest earning household member less than INR5000 | 69.8% | 70.6% |

facilitators—one each for about 20 ASHAs—and a block community mobiliser at the block level.

We trained these ASHAs to identify patients with TB symptoms during their routine work and refer them to a field coordinator (FC). The FCs further screened these patients using a symptom-based tool after obtaining their verbal consent.[10] Patients with presumptive TB identified through screening were accompanied by ASHAs to the nearest PHC for diagnostic testing and physician consultation. All patients with presumptive TB underwent sputum microscopy and chest X-ray (CXR). GeneXpert testing, if indicated by the diagnostic protocol, was conducted at the laboratory operated using project resources (figure 2). Even if CXR and sputum microscopy results were not abnormal, physicians could order a GeneXpert based on the clinical presentation. We used the standard diagnostic algorithm that is recommended by the RNTCP.[26] However, RNTCP recommendation of universal drug susceptibility testing by GeneXpert for all TB cases was being rolled out in phases and was only available in the IR as a part of the intervention.[27]

On confirmation of TB diagnosis, ASHA obtained drugs from the STS and initiated treatment at patient's residence. For each confirmed case of TB, the project paid INR200 (US$3) to ASHA for referral and INR300 (US$4.5) to ASHA for assisting in diagnosis and treatment initiation. ASHAs counselled patients on the importance of adherence and treatment completion and monitored them for adverse effects through regular follow-up household visits. They received INR400 (US$6) after their first follow-up visit and INR600 (US$9) on successful completion of treatment. In addition to these patient-focused activities, we also organised community meetings periodically to improve awareness of TB and available services under the project.

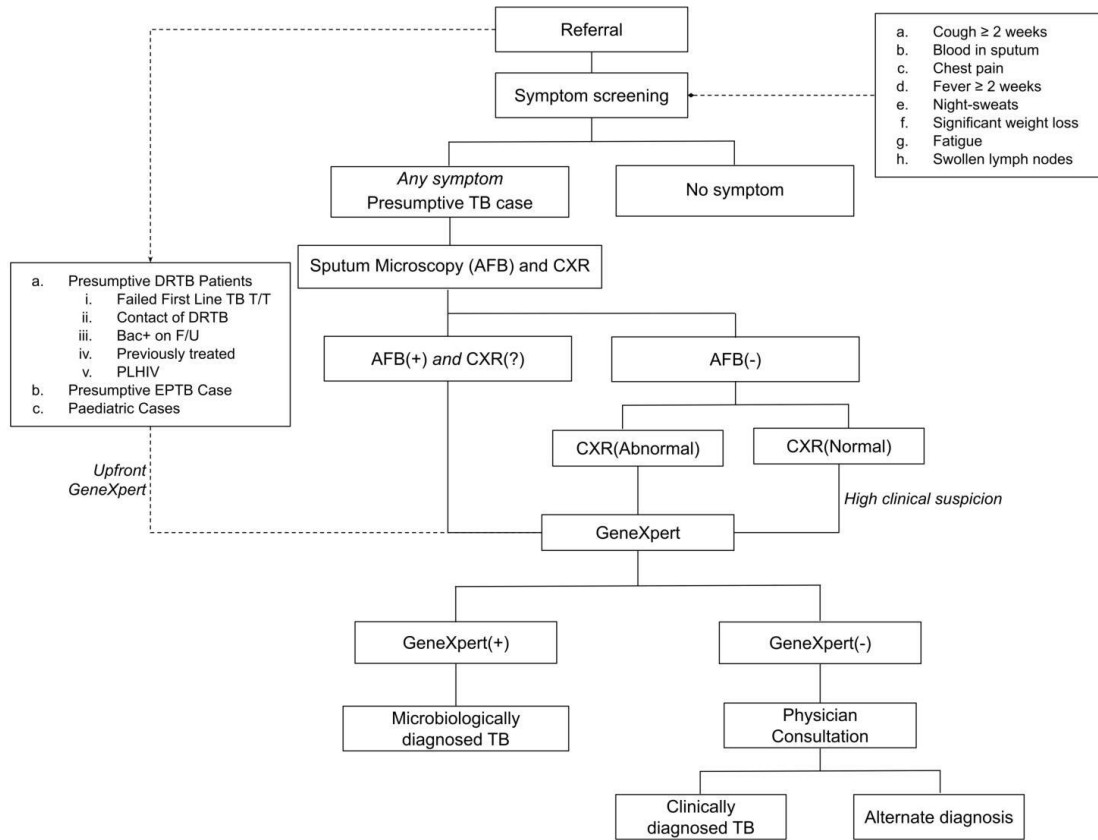

**Figure 2** The diagnostic protocol used in the active case-finding project. CXR, chest X-ray; DRTB, drug-resistant tuberculosis; DSTB, drug-sensitive TB; EPTB, extrapulmonary TB; F/U, follow-up; PLHIV, people living with HIV; T/T: treatment.

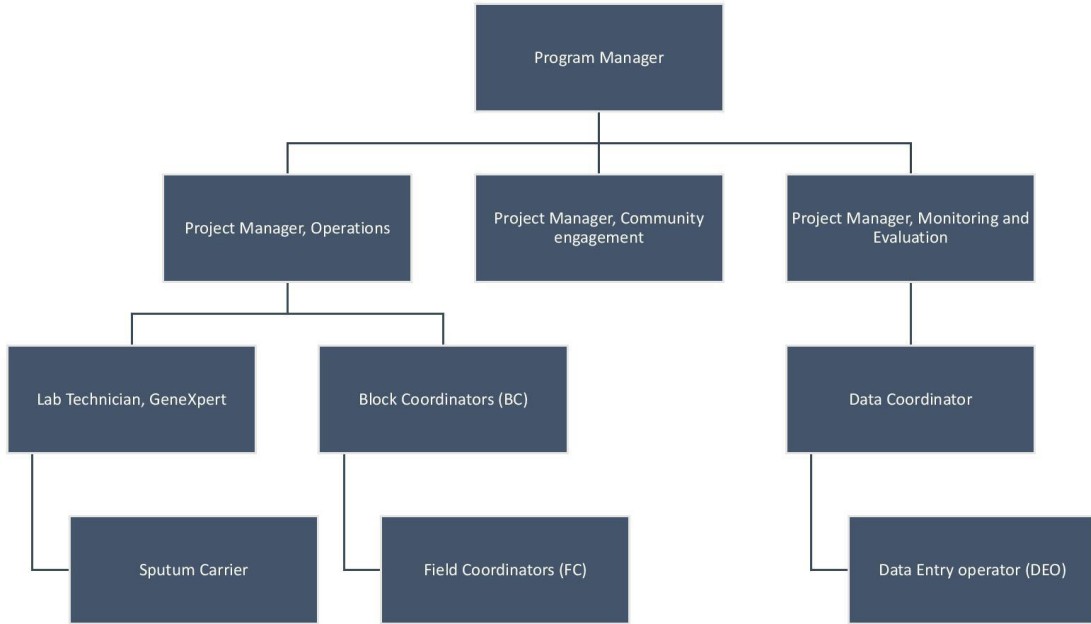

**Figure 3** The organisation chart in the active case-finding project.

The project team was led by a programme manager, who supervised three project managers responsible for operations, community engagement and monitoring and evaluation. Project manager for operations managed a team of block coordinators (BCs), one for each block in the IR, who managed a team of 6–7 FCs. Each FC covered a population of around 50 000, was responsible for training and supervision of 35–45 ASHAs, and helped with patient monitoring and community mobilisation. Supervision involved visiting patients along with ASHA, assisting the ASHAs in keeping record and filing RNTCP paperwork, and assisting ASHAs in troubleshooting across the care pathway. Also, the team included three data entry operators (DEO), data coordinator (DC), lab technician and a sputum carrier (figure 3).

### Cost framework
We used a top-down approach from the provider perspective for costing that included only costs incurred in the intervention. We defined cost efficiency in operational terms of cost per case detected to distinguish it from the more conventional term of cost-effectiveness, which is typically measured as cost per quality-adjusted life years gained or disability-adjusted life years averted.

### Data
#### Patient data
We recorded individual patient information related to referral, screening, diagnosis and treatment follow-up in paper forms. These were linked by a unique patient identifier and maintained in separate patient folders along with copies of the patient's diagnostic records. Each FC maintained folders for patients in their respective catchment areas, which were audited weekly by the BC. Trained DEOs entered data from completed forms in a patient database designed in Microsoft Excel 2016. Two DEOs checked at least one-fifth of records entered in the database for completeness and errors introduced during data entry. Besides, DC also conducted monthly audits of the patient database. Appropriate measures were taken to ensure safe-keeping of the confidential patient records.

#### Cost data
Each expense was first recorded on a paper-based voucher. A project manager verified each voucher, assigned it to one of the budget categories—staffing, activities (eg, training programmes), health commodities and services (GeneXpert, CXR), and administrative overheads—and entered the information in computer-based accounting software, Tally 11. The programme manager reconciled monthly expenses against the project budget.

#### Program data
We obtained data on quarterly TB case notifications for each block from the district programme office. We also extracted monthly data on three maternal and child health indicators representing ASHA's key activities from the NHM Health Statistics Information Portal.[28] These included the number of pregnant women registered for ANC, the number of institutional deliveries, and the number of immunisation sessions where ASHA was present. The programme data were obtained after the intervention, whereas the patient and cost data were collected in tandem with the intervention.

### Analysis
We calculated the quarterly flow of patients at each stage of the care pathway: referrals eligible for screening, patients screened, presumptive TB patients identified, patients tested, patients with confirmed TB diagnosis, and confirmed TB cases initiated on treatment. We defined the

prediagnostic lost to follow-up as the proportion of patients with presumptive TB who were not tested, and the PTLFU as the proportion of patients diagnosed with TB who were not initiated on treatment. We used the number of notified TB cases to calculate annual case notification rates per 100 000 population for IR and CR.

We calculated the quarterly averages for indicators on ASHAs' performance and mapped them to the baseline period (Q3 of 2016 to Q2 of 2017) and the study period (Q3 of 2017 to Q2 of 2018).

To calculate the intervention cost, we included all components of operational expenses (ie, excluding capital expenditure) that were incurred over and above routine programmatic activities under RNTCP. We divided these costs between case-finding and treatment categories based on actuals or the amount of time spent by the staff on the different activities estimated through semi-structured interviews. We used an exchange rate of INR67 per US$ for all costs (online supplemental figure 1).

We divided FCs' workday into three components: travel, case-finding activities and treatment support activities. We estimated the time spent on the latter two based on actual time taken for each activity per patient and average patient load per FC. We calculated travel time based on the average monthly travel reimbursement amount and allocated to it between case-finding and treatment support activities in proportion to their time spent on each of these. A similar analysis was repeated for BCs and project managers with some salient differences. We did not consider travel expenses for BCs and project managers as the amount of time spent by them on travel was minimal. The time spent by these staff members in supervision was allocated to case-finding and treatment support activities in the proportion of the time allocated by their team members on these two categories. Finally, the data management's time was divided into case-finding and treatment support categories in proportion to the total time spent by FCs, BCs, and project managers (online supplemental figure 2).

### Patient and public involvement

We neither involved patients in study design nor in the interpretation of findings.

### FINDINGS

From July 2017 to June 2018, the project received 12 394 referrals eligible for screening. Of these, 11 233 patients were screened for symptoms of TB, 9895 patients with symptoms of TB were identified. Of these, 5864 patients were tested for TB, whereas the remaining 40.7% were classified as the prediagnostic lost to follow-up. Of those tested, 1236 patients were diagnosed with TB, with 51.5% of those being confirmed with a microbiological test. Of the diagnosed patients, 1194 patients were initiated on TB treatment yielding a PTLFU of only 3.4% (figure 4) (online supplemental figure 3, online supplemental figure 4).

The notification rate in IR increased from 45.8 at baseline to 105.8 during the study period per 100 000 population but decreased from 50.7 to 45.3 in CR. Similarly, the annual notification rate per 100 000 population for microbiologically confirmed TB increased from 20.4 to 40.2 in IR but decreased from 29.3 to 22.8 in CR (table 2).

The overall average cost per diagnosed patient over the duration of the project was US$133.9, varying from a minimum of US$114 in Q3 2017 to a maximum of US$154.7 in Q4 2017. The main contributors to the cost were human resources (37.4%) and medical commodities (32.5%). Project activities and administrative overhead contributed to 20% and 10% of the cost, respectively (table 3).

The number of pregnant women registered for ANC increased by 6.1% and 3.8% in IR and CR, respectively. The number of institutional deliveries increased by 2.6% in IR as well as CR. Finally, the number of immunisation sessions where an ASHA was present increased in IR by 0.2% but decreased by 2.8% in CR (table 4).

### DISCUSSION

ACF has been widely recommended for the early identification and treatment of patients.[29] Several modelling studies in various contexts, including India, China and Uganda, have shown it to be cost-effective.[8 15 30 31] However, large-scale adoption of health interventions in resource-limited settings often requires cost efficiency in addition to cost-effectiveness. Unfortunately, there is limited and mixed evidence on cost-efficient strategies in high prevalence, resource-limited settings.[15 32] In this paper, we report on one such intervention that leveraged existing CHWs in the health system and their knowledge about community health status to drive cost efficiency. The intervention resulted in a significant increase in the notification rate at the cost of about US$134 per case diagnosed. In addition, the involvement of CHW in TB services did not adversely impact their existing tasks.

It has been suggested that leveraging existing CHWs to integrate TB screening services with other community health programmes like child immunisation can be effective.[31] However, our study is one of the first to demonstrate the practical feasibility of this approach. CHWs have extensive knowledge of the health system and are also trusted members of their communities. Consequently, they can leverage their unique position by acting as patient navigators and ensuring that they complete their pathways to treatment completion.[33–35]

Involving CHWs can also aid in engaging other actors like informal health providers and community in the way they referred people to be screened in our intervention. Their role, although, was ancillary while the FCs screened and diagnosis and treatment activities for such cases were undertaken by the CHWs.

The unit cost of our intervention was substantially lower than that of other ACF interventions in the recent past. In Cambodia, ACF strategies using CHWs report a

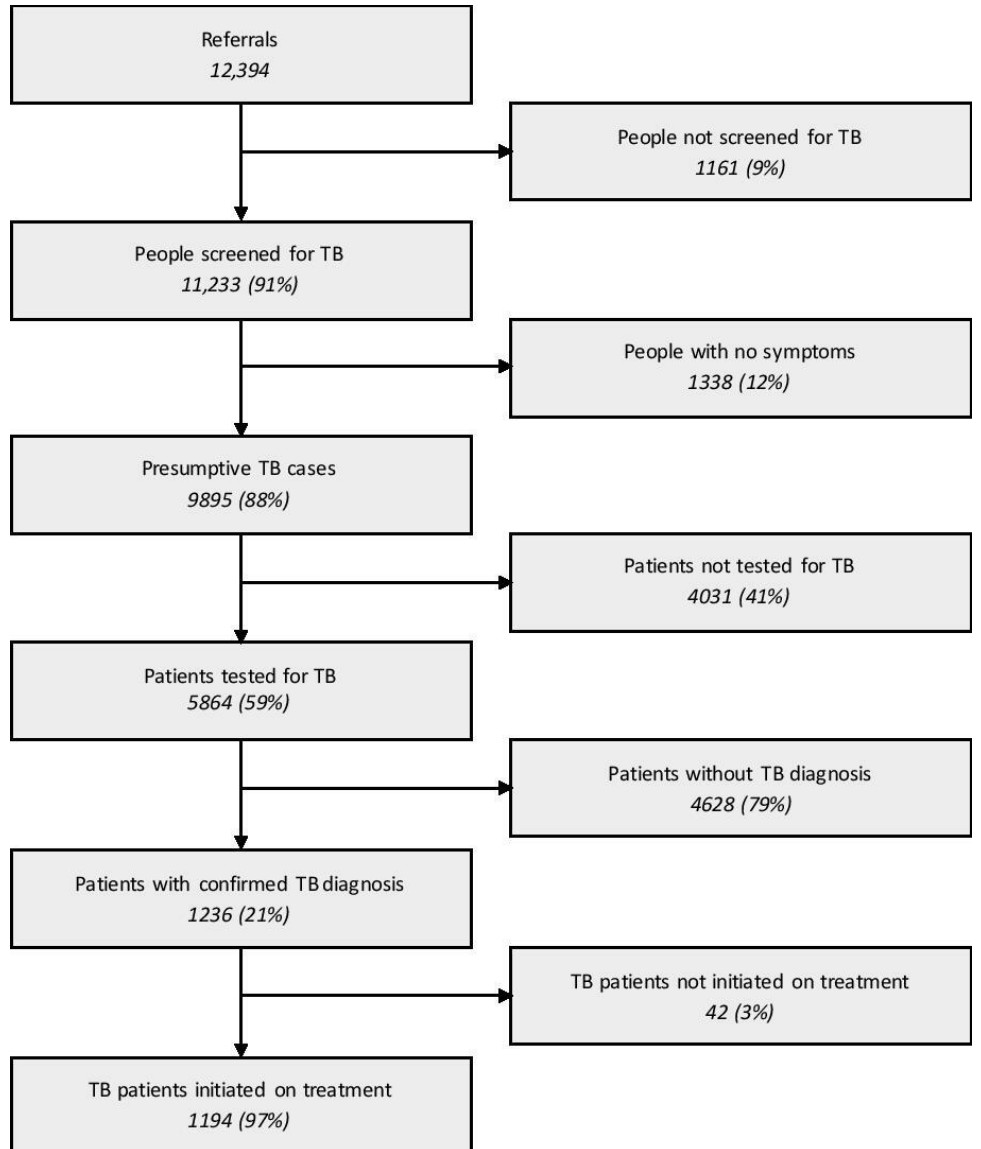

**Figure 4** The patient care cascade from Q3 2017 to Q2 2018. *All percentages are calculated as a proportion of the number of participants entering the previous step of the cascade.

cost ranging from US$249 for door-to-door screening to US$316 for symptomatic.[14] A household contact investigation strategy in urban Uganda reported a cost of US$444 per additional case diagnosed.[31] One of the main drivers for the significant cost efficiency of our intervention is that it, unlike door-to-door surveys or mass screening, relies on CHWs' experience and understanding of the community to find people at risk of TB. This approach is particularly useful and relevant in settings where TB incidence is evenly spread in the general population, and it may not be possible to target specific high-risk population segments as recommended by WHO guidelines.[10] In particular, CHWs use their own social network to filter referrals from the larger population and enrich the stream of presumptive cases compared with what would have been possible with door-to-door screening. The lower lost to follow-up, mentioned earlier, also lowers the cost per case diagnosed and initiated on treatment.[36]

Another Indian intervention that used CHWs to conduct door-to-door screening in a tribal population reported a cost of US$31 per patient, excluding drugs and diagnostics. Similar components in our intervention costed US$91 per patient. The main driver for lower cost in that intervention was a high incidence rate in the community (more than ten times the national estimate) and a smaller catchment area (approximately 1/9th of our study population), which resulted in significantly lower staffing and administrative cost.[37 38] However, the difference in costs needs to be interpreted with caution as studies vary substantially in their context (choice of ACF strategy, intervention design and diagnostic algorithm, TB epidemiology and health system characteristics) as well as their costing methodology (costing perspective (patient, provider, societal) and outcome measure).[39 40]

The key factors in explaining the efficacy of ASHAs in case finding in our intervention are the incentives and

**Table 2** TB case notification rates per 100 000 population in the public sector in the intervention and control region of the active case-finding project

| Year | Quarter | IR Bac+ | IR All cases | CR Bac+ | CR All cases |
|------|---------|---------|----------|------|-----------|
| 2016 | Q3 | 5.8 | 11.8 | 7.5 | 13.9 |
|      | Q4 | 4.3 | 9.8 | 5.7 | 11.7 |
| 2017 | Q1 | 5.4 | 11.4 | 7.6 | 12.4 |
|      | Q2 | 4.9 | 12.8 | 8.5 | 12.7 |
|      | Q3 | 7.2 | 22.3 | 6.1 | 10.2 |
|      | Q4 | 9.5 | 26 | 5.4 | 9.5 |
| 2018 | Q1 | 9.6 | 27.7 | 6 | 12.6 |
|      | Q2 | 13.9 | 29.8 | 5.3 | 13 |

*Bac+: Microbiologically confirmed TB cases.
.CR, control region; IR, intervention region; TB, tuberculosis.
.

**Table 4** ASHA's performance on reproductive, maternal and child health programme indicators in the intervention and control region in the active case-finding programme

| Indicator | | Baseline | Study period | Change, % |
|-----------|--|----------|--------------|-----------|
| No of pregnant women registered for ANC | IR | 5911 | 6270 | 6.1 |
| | CR | 6098 | 6327 | 3.8 |
| No of institutional deliveries conducted | IR | 3962 | 4065 | 2.6 |
| | CR | 3560 | 3654 | 2.6 |
| No of immunisation sessions where ASHAs were present | IR | 2550 | 2555 | 0.2 |
| | CR | 2716 | 2639 | −2.8 |

All numbers are quarterly averages.
Baseline period: Q3 of 2016 to Q2 of 2017.
Study period: Q3 of 2017 to Q2 of 2018.
ANC, antenatal checkup; ASHA, Accredited Social Health Activist; CR, control region; IR, intervention region.

the level of supervision that they received. The amount of incentive to ASHAs was competitive in comparison to their other activities, as shown above. However, its untimely disbursal is a shortcoming of the routine programmes and likely a reason ASHA partnered with our intervention, where such disbursals were prompt.[41] Further, ASHA's motivation is also dependent on her social and contextual environment, among other factors, and it was also experienced in our implementation.[42 43] ASHA's engagement level varied widely and it likely impacted the yield as well. Nonetheless, her remuneration is not considered commensurate with those of other health personnel and, overall, insufficient for the work they put in.[44–46] The drivers of engagement and the role of incentives in the poorly understood decision-making process of ASHA deserves further investigation. In a constrained health system, there are perennial concerns about overburdening CHWs with new tasks, thereby resulting in poor programme outcomes on the existing tasks.[47–49] In this context, it is encouraging that involvement of CHWs in TB ACF activities did not adversely affect their performance on tasks related to maternal and child health. Any

changes in indicators were small and of limited pragmatic significance (table 4). Our results agree with evidence from Tanzania regarding the ability of CHWs to handle multiple roles in the HIV programme as well as maternal and child health programmes. In particular, that study did not find a significant difference between the trajectory of monthly HIV visits by CHWs after they were assigned additional tasks related to maternal and child health.[48] However, any integration of CHWs in other programmes should carefully assess factors affecting their capacity and performance. In India, their training and education levels vary widely, and poor motivation and inadequate supportive supervision are well known limiting factors.[50–52]

Although the intervention produced encouraging results, there was heterogeneity in the performance metrics across the blocks, over time, and across ASHAs (online supplemental figure 4). Further efforts are needed to understand this heterogeneity better and use it for benchmarking and programme improvement. Moreover, addressing the prediagnostic lost to follow-up will likely improve the yield in such a programme. Its responsible factors are poor support at the family and health

**Table 3** Costs incurred in the active case-finding programme from Q3 of 2017 to Q2 of 2018

| Categories | 2017 Q3 | 2017 Q4 | 2018 Q1 | 2018 Q2 | Total | Proportion, % |
|-----------|---------|---------|---------|---------|-------|---------------|
| Activities | ₹433 837 | ₹501 876 | ₹620 093 | ₹666 326 | ₹2 222 132 | 20.0 |
| Administrative overheads | ₹334 235 | ₹277 192 | ₹253 173 | ₹242 279 | ₹1 106 879 | 10.0 |
| Human resources | ₹1 053 515 | ₹1167 181 | ₹934 772 | ₹996 832 | ₹4 152 300 | 37.4 |
| Commodities (drugs and diagnostics) | ₹346 683 | ₹1183 689 | ₹1 305 790 | ₹770 858 | ₹3 607 020 | 32.5 |
| Grand Total | ₹2 168 270 | ₹938 | ₹ 3 113 828 | ₹2 676 295 | ₹11 088 331 | |
| TB cases diagnosed | 284 | 302 | 324 | 326 | 1236 | |
| Cost per TB diagnosed (INR) | 7635 | 10364 | 9611 | 8209 | 8971 | |
| Cost per TB diagnosed (US$) | 114.0 | 154.7 | 143.4 | 122.5 | 133.9 | |

Exchange rate: 1 USD ($)=67 INR (₹).
TB, tuberculosis.

centre level, inadequate services in the health system, and stigma.[53] Future efforts should focus on empowerment of ASHAs and patients, and ameliorating the health system deficiencies. Its transition to a fully integrated component of the mainstream public health system is non-trivial, and past evidence of such integration, both in India and elsewhere, is mixed.[54 55] A successful transition will a require seamless interface between CHWs and senior RNCTP staff, such as the STS. During the intervention, the field team enabled this link through supportive supervision of CHWs, which is known to be a major enabler for the successful extension of CHWs' role to generate favourable outcomes.[56 57] Going forward, it would be crucial to develop a cadre of supervisors within the programme who will fulfil this function. In the absence of this supervisory capacity, each STS will have to manage 150–200 CHWs, which may not be effective. Our analysis provides a framework for calculating the cost of building this supervisory capacity, which can be incorporated in the states' annual budgeting cycles through their project implementation plan.

The main strengths of our study emanate from the fact that our intervention was a pragmatic ACF implementation that utilised existing CHWs in the health system. The study was conducted in a routine programmatic site, which simulated a typical low-resource setting environment with a regular health system. We also used routine programmatic data on case notifications for impact evaluation and other health outputs to capture any externality on the provision of other health services. We used a comparable CR within the same district to obtain the intervention's incremental effect over and above other secular changes in programme implementation. Finally, we had access to granular activity-level costing data, which limited (but did not eliminate) the need to allocate indirect costs.

However, our study also has some limitations. First, it was not designed as a randomised control trial. We purposely chose blocks in the IR based on the catchment area of the prior work done by the community-based organisation that led this intervention. The CR, though similar to the IR in many important and relevant aspects, was also purposely chosen. As a result, we cannot rigorously claim that the impact calculated from our study is caused by the intervention and is representative at the state or national level. Second, we focused only on the incremental health system cost incurred by the intervention and did not include patient costs incurred or averted as well as costs incurred by the RNTCP to coordinate with our intervention. Finally, the limited duration of our intervention did not allow us to capture longer-term health outcomes such as successful treatment completion and impact on TB epidemiology. Careful accounting of these costs and benefits would be needed to conduct a comprehensive cost-effectiveness analysis of a national scale-up of our intervention from a societal perspective.

## CONCLUSION

Leveraging existing CHWs in the health system can enhance cost efficiency of TB ACF programmes without adversely affecting the delivery of other healthcare services in their portfolio. National scale-up of this approach for TB ACF will require detailed understanding of existing capacity utilisation of CHWs due to their routine tasks and the importance of supportive supervision in helping them effectively manage the new task in addition to the routine ones.

**Acknowledgements** We acknowledge the efforts of the Accredited Health Social Activists and the project team at Innovators In Health. We presented this at the 50th Union World Conference on Lung Health 2019 and thank the audience for their comments.

**Contributors** Funding acquisition: MB, TG; Conceptualisation of the intervention: TG, MB; Conceptualisation of the analysis: TG, SD; Data collection: TG; Data analysis and interpretation: TG, SD; Writing-original draft: TG; Writing-review and edits: TG, SD and MB.

**Funding** This project was supported by the Stop TB Partnership's TB REACH initiative and was funded by the Government of Canada and the Bill & Melinda Gates Foundation. Manish Bhardwaj was supported by a Grand Challenges Explorations grant number OPP1190735 from the Bill & Melinda Gates Foundation.

**Disclaimer** Neither funders had any role in the study design, data collection and analysis, decision to publish, or preparation of the manuscript.

**Map disclaimer** The depiction of boundaries on this map does not imply the expression of any opinion whatsoever on the part of BMJ (or any member of its group) concerning the legal status of any country, territory, jurisdiction or area or of its authorities. This map is provided without any warranty of any kind, either express or implied.

**Competing interests** None declared.

**Patient and public involvement** Patients and/or the public were not involved in the design, or conduct, or reporting, or dissemination plans of this research.

**Patient consent for publication** Not required.

**Ethics approval** Ethical approval was obtained from the Institutional Review Board at Indian School of Business, Hyderabad. The board waived the informed consent requirement for the study. Further, only aggregate intervention data was used for the analysis.

**Provenance and peer review** Not commissioned; externally peer reviewed.

**Data availability statement** All data relevant to the study are included in the article or uploaded as online supplemental information. The cost, programme, and yield data used in the study are available in online supplemental files 1, 3, 4, respectively.

**ORCID iDs**
Tushar Garg http://orcid.org/0000-0002-6781-8574
Manish Bhardwaj http://orcid.org/0000-0001-7885-8005
Sarang Deo http://orcid.org/0000-0002-3233-6014

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
