## [Reviewer comments · BMJ Open]

ARTICLE DETAILS

TITLE (PROVISIONAL)	Role of community health workers in improving cost efficiency in an active case finding tuberculosis program: An operational research study from rural Bihar, India
AUTHORS	Garg, Tushar; Bhardwaj, Manish; Deo, Sarang

VERSION 1 – REVIEW

REVIEWER	Kelsey Vaughan Bang for Buck Consulting, The Netherlands
REVIEW RETURNED	09-Jan-2020

GENERAL COMMENTS	This is an interesting study and a well-written manuscript. I have only a few minor revisions to improve clarity: - The abstract notes program dates of May 2017 to June 2018 but the methods note July 2017 to June 2018; please clarify- You should clarify up front what you mean by 'cost-effective' (based on WHO GDP/capita thresholds which are now largely discredited?) and 'cost-efficient'. These terms are frequently misused and therefore are not understood in the same way by everyone- Am I correct that data collect occurred simultaneously during the program period? Ie as you implemented the intervention you tracked patient data, cost data and the program data?- The methods do not explicitly specify the costing perspective taken; it seems it's a provider perspective, but please clarify- The methods do not explicitly state that purposive sampling was used to select the control region. Was proximity but separation the only variable used in the purposive sampling? Please rewrite. Table 1 gives the impression the intervention and control regions are comparable but then the findings in terms of number of immunization sessions where an ASHA (which increased in the IR but decreased in the CR) made me question if there wasn't some other intervention, program or characteristic that distinguished the regions- The finding of a decreased number of immunization sessions in the CR is surprising; please comment on this in the discussion- The discussion should note in which contexts modelling exercises have show ACF to be cost-effective; was this in India or somewhere else?- You nicely outline the cost of other ACF interventions but were the methods used to arrive at these estimates comparable to your study? This is a common problem in costing and should be checked, and any major methodological differences noted- Your discussion does not show how the intervention is cost-efficient, though you note in your conclusion that it is
--

REVIEWER	Luan Vo Friends for International TB Relief, Viet Nam
REVIEW RETURNED	08-Mar-2020

GENERAL COMMENTS	Overall assessment This was a reasonably well-written manuscript for a study with an incredibly high yield and compelling cost argument. It strongly suggests the expansion of community engagement for TB case finding, which is in line with WHO End TB strategy recommendations. The attempt to assess the change in activity level of the ASHA's other responsibilities was particularly noteworthy and commendable. There are several areas on which the authors should expound. In the methods this request particularly relates to ethics, patient consent and confidentiality. The yield calculations and denominator used should also be further explained given the dramatically low number needed to screen. The authors should also consider a more comprehensive cost analysis inclusive of how much of their output is attributable to the project funds compared to RNTCP's core investment. The impact assessment of the additional TB screening activities on ASHA workload could have been more rigorous as top-down indicator analyses may have masked individual impacts for those who truly committed time to the TB case finding work. In general, a deeper engagement with the ASHAs, enabling factors and barriers of their engagement would have been highly welcomed. Another critical component in need of expansion is the role of the supervisors. There was insufficient detail around the role as enablers of success and the challenges they faced in the methods and results. As such, the discussion and conclusion around this part of the study appear insufficiently substantiated. Nevertheless, overall the manuscript contains interesting findings that promote the evidence base on TB case finding using community networks as per WHO recommendations. The manuscript is composed at an acceptable scientific and linguistic standard. As such, it may be considered for publication once the issues below are addressed to the satisfaction of the editor(s) and journal. Summary The study was a controlled, quasi-experimental cohort study to evaluate the use of ASHAs for integrated TB case finding. The study was implemented for one year in one district of Bihar, India. Key outcome measures included yield, changes in notification and cost per case detected. Abstract - Please explain acronyms on the first use in the abstract and then again in the manuscript. Background - Please provide a source that offers evidence of the linkage between ACF and decrease in incidence (rows 19-22, page 1) as the sources you have cited may be insufficient to evince a clear linkage between ACF and long-term reduction of incidence or population-level impact. There are also publications that have similarly stated the need for more evidence around this topic.^{1,2} As such, you may want to consider providing additional evidence in support of the assertion that ACF results in incidence reduction or consider rephrasing. - Please provide more information on the ASHA's that delineates
---

their education level, amount of time committed to current responsibilities, remuneration for these standard ASHA activities and supervisory structures. This will be helpful to contextualize the impact assessment of the additional TB case finding activities on the ASHA's existing workload.

Methods

- In table 1 please provide information on the TB burden in those districts and an estimate of the number of missed cases or any information available around local prevalence. Please also provide the number of ASHAs in the IR and CR and other quantifiable, relevant information regarding these ASHAs.
- Please add a sentence or two comparing the study's diagnostic algorithm with that of the RNTCP.
- In the methods section, please provide an explanation and/or example of "high clinical suspicion" that would warrant a GeneXpert test for someone with a normal chest X-ray as shown in your diagnostic algorithm.
- Please include a statement on ethical approvals, patient consent and appropriate management and safe-keeping of confidential patient records in this section. While you state the use of aggregate data as the basis for exemption of ethical approvals, the methods describe the collection of individual records and data that were used to construct the TB care cascade. As such you may want to consider obtaining a retrospective consent waiver from an ethics committee.
- Please define the difference between cost-effective and cost-efficient in this manuscript (and specifically mentioned in the discussion).

Results

- Please verify that the denominators are complete, since presumptive TB cases comprise 79.7% of referrals and 88.1% persons screened for TB. That is a very high rate of symptomatic persons encountered in the community. Moreover, the implied number needed to screen is 9, which is lower than household contacts and persons living with HIV.³
- Please provide an analysis and discussion on the number of ASHAs in the intervention area and the number of people referred/screened. Based on the information provided in the manuscript (6-7 FCs responsible for 45k people and 50 ASHAs each), there may have been around 300 ASHAs in the intervention region, which means each ASHA on average referred 3 person per month. Please present the ASHA's capacity utilization and address it – whether optimal or not – in the discussion.
- Please provide a breakdown of the TB care cascade or at least the yield in the IR by block to see any heterogeneity in your implementation.

Discussion

- It would be very interesting to detail and discuss the current level of time spent by ASHAs on their regular duties since the conclusion is that the addition of TB screening did not affect their ongoing activities.
- Please also comment on the current compensation scheme (fixed and variable) for ASHAs to fulfill their other duties and compare that with the incentives you have provided for TB case finding on this project. Were they competitively powered? If so, show and discuss how they compared.
- Please include a discussion on the pre-diagnostic loss to follow-up (>40%), reasons and potential future mitigation efforts that would

	leverage the ASHAs or existing health system (e.g., sputum collection and transport).  - It is not advisable to compare cost per case detected across different contexts. Projects in Cambodia and Uganda face different sociocultural, institutional and economic barriers as well as routine program performance and health system infrastructure that may affect costs to find an additional case. From a philosophical perspective, it is also not helpful to engage in a race to the bottom with respect to cost per case detected as the marginal cost for every additional case detected will invariably rise as we close the case finding gap. - The discussion on lines 8-25 on page 14 is highly appropriate and on point, but seems insufficiently substantiated based on the prior content of the background, methods or results. As alluded to in earlier comments, it would be highly advisable to provide more information and description of the ASHAs to make the point here about the supervisory shortcomings of the existing system. - None of the sources cited on line 5, page 15 measured or provided evidence of a reduction in TB incidence, so it may be advisable to rephrase the sentence. Conclusion  - There was insufficient evidence presented in the results and in the manuscript overall about the effectiveness of supervision, so that the comment here seems insufficiently substantiated. Please consider revising and drawing conclusions based on the presented and discussed results or expanding the results to emphasize the role of supervision on this study. References  1. Koura KG, Trébucq A, Schwoebel V. Do active case-finding projects increase the number of tuberculosis cases notified at national level? Int J Tuberc Lung Dis. 2017;21(1):73–8. 2. Kranzer K, Afnan-Holmes H, Tomlin K, Golub JE, Shapiro AE, Schaap A, et al. The benefits to communities and individuals of screening for active tuberculosis disease: A systematic review. Int J Tuberc Lung Dis. 2013;17(4):432–46. 3. Shapiro A, Akande T, Lonnroth K, Golub J, Chakravorty R. A systematic review of the number needed to screen to detect a case of active tuberculosis in different risk groups [Internet]. WHO TB Review. 2013. Available from: http://www.who.int/tb/Review3NNS_case_active_TB_riskgroups.pdf
--	---

VERSION 1 – AUTHOR RESPONSE

Reviewer(s)' Comments to Author:

Reviewer: 1

Reviewer Name: Kelsey Vaughan

Institution and Country: Bang for Buck Consulting, The Netherlands

Please state any competing interests or state 'None declared': None declared

Please leave your comments for the authors below:

This is an interesting study and a well-written manuscript. I have only a few minor revisions to improve clarity.

Thank you for your valuable comments.

- The abstract notes program dates of May 2017 to June 2018 but the methods note July 2017 to June 2018; please clarify

The first two months of the program, i.e. May and June 2017, were preparatory and spent in activities like recruitment, developing partnerships, setting protocols etc. The fully fledged program operations started in July 2017. We have added text that clearly states this difference on Page 5 under Study Design.

- You should clarify up front what you mean by 'cost-effective' (based on WHO GDP/capita thresholds which are now largely discredited?) and 'cost-efficient'. These terms are frequently misused and therefore are not understood in the same way by everyone Thank you for the observation. Indeed, these terms have multiple interpretations in the literature, which can lead to confusion for the reader especially, if both are used simultaneously in the manuscript. Further, we acknowledge your concern on using GDP per capita threshold but are unsure if the thresholds are completely abandoned and discredited#. The studies we refer to also use this threshold to establish the cost-effectiveness. Hence, we cautiously suggest that modeling studies have shown ACF to be cost-effective.

We have added the following in the Methods section to clear this confusion from the manuscript:

Cost framework

We used a top-down approach from the provider perspective for costing that included only costs incurred in the intervention. We defined cost-efficiency in operational terms of cost per case detected to distinguish it from the more conventional term of cost-effectiveness, which is typically measured as cost per QALY (quality-adjusted life years) gained or DALY (disability-adjusted life years) averted.

#Azman AS, Golub JE, Dowdy DW. How much is tuberculosis screening worth? Estimating the value of active case finding for tuberculosis in South Africa, China, and India. BMC Medicine 2014;:9. doi:[10/gb33tt](https://doi.org/10/gb33tt);

#Dobler CC. Screening strategies for active tuberculosis: focus on cost-effectiveness. ClinicoEconomics and Outcomes Research 2016;8:335–47. doi:[10/gdxtmw](https://doi.org/10/gdxtmw);

#Lung T, Marks GB, Nhung NV, et al. Household contact investigation for the detection of tuberculosis in Vietnam: economic evaluation of a cluster-randomised trial. *The Lancet Global Health* 2019;7:e376–84. doi:[10/gf5tdz](https://doi.org/10.1016/S2468-2667(19)30512-1).

- Am I correct that data collect occurred simultaneously during the program period? Ie as you implemented the intervention you tracked patient data, cost data and the program data?

Indeed, the data collection for all but one dataset was in tandem with the intervention. The data on maternal and child health indicators representing ASHAs' key activities were extracted from the National Health Mission's Health Statistics Information Portal in February 2019.

We've added the following under Data in the Methods section:

The program data was obtained after the intervention, whereas the patient and cost data were collected in tandem with the intervention.

- The methods do not explicitly specify the costing perspective taken; it seems it's a provider perspective, but please clarify

Indeed, we have taken a provider perspective. This has been added in the Methods section under Cost Framework. Thank you for raising this.

- The methods do not explicitly state that purposive sampling was used to select the control region. Was proximity but separation the only variable used in the purposive sampling? Please rewrite. Table 1 gives the impression the intervention and control regions are comparable but then the findings in terms of number of immunization sessions where an ASHA (which increased in the IR but decreased in the CR) made me question if there wasn't some other intervention, program or characteristic that distinguished the regions.

The choice of control region was based on similar geography, population, sociodemographic characteristics, health system characteristics, and TB epidemiology. We have clearly mentioned this criteria under Study Design now.

The health system characteristics that we relied on included health infrastructure, availability of health services, human resources, operational synchrony etc. Since the health system governance of IR and CR were under the same district administration, the overall decisions applied equally to all the blocks. Further, comparing data on health outcomes like IMR, MMR etc. would have been ideal but these data, unfortunately, were not available at the block level.

- The finding of a decreased number of immunization sessions in the CR is surprising; please comment on this in the discussion

We understand your concern but please note that there are several reasons why this may not be a serious issue. The indicator in question—number of immunization sessions where an ASHA was present—is not necessarily equal to the number of immunization sessions held. These sessions are

led by Auxiliary Nurse Midwife while ASHAs helps in mobilizing mothers and children and maintaining records. Further, such a small difference (-2.8%) [n=77] in quarterly averages has limited pragmatic significance. In any case, we mention it in the Discussion section as per the suggestion of the reviewer.

In this context, it is encouraging that involvement of CHWs in TB ACF activities did not adversely affect their performance on tasks related to maternal and child health. Any changes in indicators were small and of limited pragmatic significance.

- The discussion should note in which contexts modelling exercises have shown ACF to be cost-effective; was this in India or somewhere else?

We agree with the reviewer and have accordingly added the countries where the modelling studies were based. We mention it in the Discussion section now:

Several modelling studies in various contexts, including India, China, and Uganda, have shown it to be cost-effective

- You nicely outline the cost of other ACF interventions but were the methods used to arrive at these estimates comparable to your study? This is a common problem in costing and should be checked, and any major methodological differences noted.

We thank the reviewer for raising this important issue, which has also been recognized by scholars and experts. Based on the suggestion, we have added the following description in the Discussion section:

However, the difference in costs needs to be interpreted with caution as studies vary substantially in their context (choice of ACF strategy, intervention design and diagnostic algorithm, TB epidemiology, and health system characteristics) as well as their costing methodology (costing perspective (patient, provider, societal) and outcome measure).

- Your discussion does not show how the intervention is cost-efficient, though you note in your conclusion that it is.

Thank you for your comment. In the revised text, we compare the costs in our study with those in other ACF programs. We roughly use cost-efficiency to mean lower cost per detected case as mentioned above. Based on this, we mention in the Discussion section now that “the unit cost of our intervention was substantially lower than that of other ACF interventions in the recent past.” Comparing costs with that of active and passive finding in the Revised National TB Control Program (RNTCP) would have been ideal but such evidence isn’t yet available to the best of our knowledge. While a study does estimate unit costs in RNTCP, its focus is cost to provider of treating a confirmed TB patient.[§]

We agree with your earlier comment on challenges in comparison of costs. There is paucity of data from active case finding TB programs, and difference in service delivery approach and costing methodology raises further challenges. Hence, our conclusion of cost-efficiency need to be taken with caution.

§ Muniyandi, M and Rajeswari, R and Balasubramanian, R (2006) Estimating provider cost for treating patients with tuberculosis under Revised National Tuberculosis Control Programme (RNTCP). Indian Journal of Tuberculosis, 53 (1). pp. 12-17. ISSN 0019-5705

Reviewer: 2

Reviewer Name: Luan Vo

Institution and Country: Friends for International TB Relief, Viet Nam

Please state any competing interests or state 'None declared': None declared

Please leave your comments for the authors below:

Overall assessment

This was a reasonably well-written manuscript for a study with an incredibly high yield and compelling cost argument. It strongly suggests the expansion of community engagement for TB case finding, which is in line with WHO End TB strategy recommendations. The attempt to assess the change in activity level of the ASHA's other responsibilities was particularly noteworthy and commendable.

There are several areas on which the authors should expound. In the methods this request particularly relates to ethics, patient consent and confidentiality. The yield calculations and denominator used should also be further explained given the dramatically low number needed to screen. The authors should also consider a more comprehensive cost analysis inclusive of how much of their output is attributable to the project funds compared to RNTCP's core investment.

The impact assessment of the additional TB screening activities on ASHA workload could have been more rigorous as top-down indicator analyses may have masked individual impacts for those who truly committed time to the TB case finding work. In general, a deeper engagement with the ASHAs, enabling factors and barriers of their engagement would have been highly welcomed. Another critical component in need of expansion is the role of the supervisors. There was insufficient detail around the role as enablers of success and the challenges they faced in the methods and results. As such, the discussion and conclusion around this part of the study appear insufficiently substantiated.

Nevertheless, overall the manuscript contains interesting findings that promote the evidence base on TB case finding using community networks as per WHO recommendations. The manuscript is composed at an acceptable scientific and linguistic standard. As such, it may be considered for publication once the issues below are addressed to the satisfaction of the editor(s) and journal.

Thank you for your comments. We've significantly expanded on ASHA in both the Methods and the Discussion section. It now includes various details on their selection criteria, remuneration, and supervisory structure. In addition, the findings are complemented with discussion on their capacity, supportive supervision, and incentive. We appreciate your suggestion to further investigate factors

affecting ASHAs' engagement, understanding supervisors' role in the intervention, and include RNTCP's investment in the cost analysis. We have incorporated this in the discussion and limitations sections, and hope to undertake such investigation in the future.

Summary

The study was a controlled, quasi-experimental cohort study to evaluate the use of ASHAs for integrated TB case finding. The study was implemented for one year in one district of Bihar, India. Key outcome measures included yield, changes in notification and cost per case detected.

Abstract

- Please explain acronyms on the first use in the abstract and then again in the manuscript.

We've expanded the acronyms as suggested. Thank you for the comment.

Background

- Please provide a source that offers evidence of the linkage between ACF and decrease in incidence (rows 19-22, page 1) as the sources you have cited may be insufficient to evince a clear linkage between ACF and long-term reduction of incidence or population-level impact. There are also publications that have similarly stated the need for more evidence around this topic.^{1,2} As such, you may want to consider providing additional evidence in support of the assertion that ACF results in incidence reduction or consider rephrasing.

We agree with your observation. Although the evidence of ACF's impact on TB epidemiology is inconclusive, modelling studies have shown plausible reduction in incidence. We've rephrased the assertion and added a new reference.

Active case-finding (ACF) can address these challenges by finding previously undetected cases and promptly initiating them on treatment. Modeling studies estimate that such strategies can decrease TB incidence

Dowdy DW, Lotia I, Azman AS, et al. Population-Level Impact of Active Tuberculosis Case Finding in an Asian Megacity. PLoS ONE 2013;8:e77517. doi:[10/gcz84s](https://doi.org/10.1371/journal.pone.0077517)

- Please provide more information on the ASHA's that delineates their education level, amount of time committed to current responsibilities, remuneration for these standard ASHA activities and supervisory structures. This will be helpful to contextualize the impact assessment of the additional TB case finding activities on the ASHA's existing workload. Thank you for your comment. We've added this information in the Methods section under the Intervention heading.

Methods

- In table 1 please provide information on the TB burden in those districts and an estimate of the number of missed cases or any information available around local prevalence. Please also provide the number of ASHAs in the IR and CR and other quantifiable, relevant information regarding these ASHAs.

Thank you for your comment. Unfortunately, prevalence data is available neither at the district nor at the sub-district level. Similarly, it is not possible to estimate the number of missing cases in the IR and CR. The case notification rate is the best available indicator at present. Further, there is little quantifiable information available on ASHAs beyond their number in the IR and performance indicators mentioned in Table 4 of the findings section.

Additional information on ASHA has also been presented in Supplementary File 3 now.

- Please add a sentence or two comparing the study's diagnostic algorithm with that of the RNTCP.

We thank the reviewer for the suggestion. "We used the standard diagnostic algorithm that is recommended by the RNTCP. However, RNTCP recommendation of universal drug susceptibility testing by GeneXpert for all TB cases was being rolled out in phases and was only available in the IR as a part of the intervention." These lines have also been added in the Methods section.

- In the methods section, please provide an explanation and/or example of "high clinical suspicion" that would warrant a GeneXpert test for someone with a normal chest X-ray as shown in your diagnostic algorithm.

Thank you for your suggestion. This step in the diagnostic algorithm meant that "even if CXR and sputum microscopy results were not abnormal, physicians could order a GeneXpert based on the clinical presentation." Further, this step was congruent with RNTCP's algorithm. We've added it in the Methods section now.

- Please include a statement on ethical approvals, patient consent and appropriate management and safe-keeping of confidential patient records in this section. While you state the use of aggregate data as the basis for exemption of ethical approvals, the methods describe the collection of individual records and data that were used to construct the TB care cascade. As such you may want to consider obtaining a retrospective consent waiver from an ethics committee.

Thank you for the suggestion. We have obtained an approval and consent waiver from an IRB, and mention the following in the manuscript: "Ethical approval was obtained from the

Institutional Review Board at Indian School of Business, Hyderabad. The board waived the informed consent requirement for the study. Further, only aggregate intervention data was used for the analysis."

As suggested, we've also included that "appropriate measures were taken to ensure safe-keeping of the confidential patient records" and that "the FCs further screened these patients using a symptom-based tool after obtaining their verbal consent."

- Please define the difference between cost-effective and cost-efficient in this manuscript (and specifically mentioned in the discussion).

Thank you for your comment. “We defined cost-efficiency in operational terms of cost per case detected to distinguish it from the more conventional term of cost-effectiveness, which is typically measured as cost per QALY (quality-adjusted life years) gained or DALY (disability-adjusted life years) averted.” We’ve added this under Cost Framework sub-heading in the Methods section.

Results

- Please verify that the denominators are complete, since presumptive TB cases comprise 79.7% of referrals and 88.1% persons screened for TB. That is a very high rate of symptomatic persons encountered in the community. Moreover, the implied number needed to screen is 9, which is lower than household contacts and persons living with HIV.³ We understand your concern about the seemingly high proportion of symptomatic persons encountered in the community. It is a result of the *referral* process, which is one of the key components of the intervention. In particular, the intervention relies on ASHAs’ knowledge about the health of the community members.

They intuitively/informally screen people in the community while they are at work and register only a subset of these people as “referrals or people eligible for screening”. For example: an ASHA will meet the same set of people multiple times a week for different activities like immunization, antenatal and postnatal care, eligible couple and other surveys etc. She will ask after their family’s health and also seek names of symptomatic people in their family or neighbourhood. In other instances, people will also reach out after they know that ASHA can help patients with such symptoms. This is a subjective process and a second level of screening by the field coordinators (FCs) ensures its accuracy. The FCs used a paper-based questionnaire to screen the *referrals* to identify the presumptive TB cases.

As you correctly suggest, the denominator for these proportions should be much higher and, hence, NNS a lot smaller. However, it is not possible to assess the true number of people continually screened by ASHAs during their regular work before they are entered in the program registers.

We would like to note that this unique process of referral that leverages the community knowledge of the ASHA is the key driver of cost-efficiency in this model and, we believe, paves the way for integrating ACF in the national TB program in a practical manner.

- Please provide an analysis and discussion on the number of ASHAs in the intervention area and the number of people referred/screened. Based on the information provided in the manuscript (6-7 FCs responsible for 45k people and 50 ASHAs each), there may have been around 300 ASHAs in the intervention region, which means each ASHA on average referred 3 person per month. Please present the ASHA’s capacity utilization and address it – whether optimal or not – in the discussion.

Thank you for your comment. There were 793 ASHAs in the intervention region who referred 12,394 people eligible for screening. It averages to around 16 per ASHA per year. However, our implementation experience indicates not all ASHAs were equally involved. Although it is important to understand this distribution and reasons affecting ASHAs’ participation in the intervention, a detailed analysis is beyond the scope and objectives of this manuscript. We mention this issue in the discussion section and invite future work in this direction.

We’ve also presented this information clearly in supplementary file 3 now along with other information on ASHAs. We had earlier made an error in the methods section which has now been corrected to

“each FC covered a population of around 50,000 (45,000), was responsible for training and supervision of 35-45 (40-50) ASHAs...”

We have included the following in the Discussion section:

...ASHA's motivation is also dependent on her social and contextual environment amongst other factors and it was also experienced in our implementation. ASHA's engagement level varied widely and likely impacted the yield as well.

...any integration of CHWs in other programs should carefully assess factors affecting their capacity and performance. In India, their training and education levels vary widely, and poor motivation and inadequate supportive supervision are well known limiting factors.

- Please provide a breakdown of the TB care cascade or at least the yield in the IR by block to see any heterogeneity in your implementation.

Thank you for your suggestion. We have presented the disaggregated data by quarter and block in Supplementary file 4.

Indeed, there was heterogeneity across blocks and quarters. Notable were pre-treatment loss to follow-up, proportion of patients diagnosed with TB amongst tested, and proportion of microbiologically-confirmed cases. Further efforts are needed to better understand this.

We've added the following in the discussion section:

There was heterogeneity in the performance metrics across the blocks, over time, and across ASHAs.[Supplementary file 4] Further efforts are needed to better understand this heterogeneity and use it for benchmarking and program improvement

Discussion

- It would be very interesting to detail and discuss the current level of time spent by ASHAs on their regular duties since the conclusion is that the addition of TB screening did not affect their ongoing activities.

Thank you for your comment. Unfortunately, collecting such data is effort intensive and we did not rigorously collect it during the course of the intervention. This is the reason we used output metrics directly to argue that their involvement in our program did not affect their routine work.

There are opposing views on capacity utilization of ASHAs. On one hand, some studies claim that they are overworked whereas on the other hand, there is anecdotal evidence that they have sufficient spare time to undertake more activities. Further, this may differ across ASHAs in the same geographic location. We agree that more formal studies are needed to ascertain this heterogeneity in capacity utilization of ASHAs and their ability to absorb more tasks.

- Please also comment on the current compensation scheme (fixed and variable) for ASHAs to fulfill their other duties and compare that with the incentives you have provided for TB case finding on this project. Were they competitively powered? If so, show and discuss how they compared.

Thank you, we've detailed their current compensation in the Methods and discuss it as follows in the revised text:

The amount of incentive to ASHAs was competitive in comparison to their other activities, as shown above. However, its untimely disbursement is a shortcoming of the routine programs and likely a reason ASHA partnered with our intervention, where such disbursements were prompt. Further, ASHA's motivation is also dependent on her social and contextual environment, amongst other factors, and it was also experienced in our implementation. ASHA's engagement level varied widely and likely impacted the yield as well. Nonetheless, her remuneration is not considered commensurate with those of other health personnel and, overall, insufficient for the work they put in. The drivers of engagement and the role of incentives in the poorly understood decision-making process of ASHA deserves further investigation.

- Please include a discussion on the pre-diagnostic loss to follow-up (>40%), reasons and potential future mitigation efforts that would leverage the ASHAs or existing health system (e.g., sputum collection and transport).

Predagnostic loss to follow-up resulted from poor support at the family and health centre level, inadequate services in the health system, stigma, and logistical challenges. Patient accompaniment by ASHA and her guidance helped the patients in accessing the services. Future efforts should focus on empowerment of ASHAs and patients, and ameliorating the health system deficiencies to enable access to care.

The detailed investigation has been presented in the following mixed-methods study:

Garg T, Gupta V, Sen D, et al. Prediagnostic loss to follow-up in an active case finding tuberculosis programme:

a mixed-methods study from rural Bihar, India. *BMJ Open* 2020;10:e033706.
doi:[10.1136/bmjopen-2019-033706](https://doi.org/10.1136/bmjopen-2019-033706)

- It is not advisable to compare cost per case detected across different contexts. Projects in Cambodia and Uganda face different sociocultural, institutional and economic barriers as well as routine program performance and health system infrastructure that may affect costs to find an additional case. From a philosophical perspective, it is also not helpful to engage in a race to the bottom with respect to cost per case detected as the marginal cost for every additional case detected will invariably rise as we close the case finding gap. Thank you for raising this. We agree with your comment. We've added the following in the Discussion section:

However, the difference in costs needs to be interpreted with caution as studies vary substantially in their context (choice of ACF strategy, intervention design and diagnostic algorithm, TB epidemiology, and health system characteristics) as well as their costing methodology (costing perspective (patient, provider, societal) and outcome measure).

- The discussion on lines 8-25 on page 14 is highly appropriate and on point, but seems insufficiently substantiated based on the prior content of the background, methods or results. As alluded to in earlier comments, it would be highly advisable to provide more information and description of the ASHAs to make the point here about the supervisory shortcomings of the existing system.

Thank you, we've added the information on ASHA in the Methods section as detailed above.

- None of the sources cited on line 5, page 15 measured or provided evidence of a reduction in TB incidence, so it may be advisable to rephrase the sentence.

Thank you, we've rephrased it to the following: "Finally, limited duration of our intervention did not allow us to capture longer-term health outcomes such as successful treatment completion and impact on TB epidemiology".

Conclusion

- There was insufficient evidence presented in the results and in the manuscript overall about the effectiveness of supervision, so that the comment here seems insufficiently substantiated. Please consider revising and drawing conclusions based on the presented and discussed results or expanding the results to emphasize the role of supervision on this study.

Thank you, we agree with your comment that we do not have sufficiently rigorous and comprehensive evidence on the impact of supportive supervision on the results of our study. We have accordingly revised the conclusion to exclude this. In particular, we write that "leveraging existing CHWs in the health system can enhance cost-efficiency of tuberculosis active case finding programs without adversely affecting the delivery of other healthcare services in their portfolio. National scale-up of this approach for tuberculosis active case finding will require detailed understanding of existing capacity utilization of CHWs due to their routine tasks and the importance of supportive supervision in helping them effectively manage the new task in addition to the routine ones."

References

1. Koura KG, Trébuq A, Schwoebel V. Do active case-finding projects increase the number of tuberculosis cases notified at national level? *Int J Tuberc Lung Dis.* 2017;21(1):73–8.
2. Kranzer K, Afnan-Holmes H, Tomlin K, Golub JE, Shapiro AE, Schaap A, et al. The benefits to communities and individuals of screening for active tuberculosis disease: A systematic review. *Int J Tuberc Lung Dis.* 2013;17(4):432–46.
3. Shapiro A, Akande T, Lonroth K, Golub J, Chakravorty R. A systematic review of the number needed to screen to detect a case of active tuberculosis in different risk groups [Internet]. WHO TB Review. 2013. Available from: http://www.who.int/tb/Review3NNS_case_active_TB_riskgroups.pdf

VERSION 2 – REVIEW

REVIEWER	Luan Vo Friends for International TB Relief, Viet Nam Interactive Research and Development, Viet Nam
REVIEW RETURNED	13-Jul-2020
GENERAL COMMENTS	I would like to commend the authors for their thorough and appropriate engagement with the feedback and congratulate them on the success of their project and their contribution to the body of evidence on the effective use of CHWs in TB ACF.